**Data Availability Statement:** The reasons for the data restriction include concerns for breaching

# "We only trust each other": A qualitative study exploring the overdose risk environment among persons who inject drugs living with HIV in Nairobi, Kenya

Megan Maurano[1]*, David Bukusi[2], Sarah Masyuko[3,4], Rose Bosire[5], Esther Gitau[6], Brandon L. Guthrie[4,7], Aliza Monroe-Wise[4], Helgar Musyoki[3], Mercy Apiyo Owuor[8], Betsy Sambai[8], William Sinkele[6], Hanley Kingston[4], Carey Farquhar[1,4,7], Loice Mbogo[8‡], Natasha T. Ludwig-Barron[4,7‡]

1 Department of Medicine, University of Washington School of Medicine, Seattle, Washington, United States of America, 2 VCT and HIV Care, Kenyatta National Hospital, Nairobi, Kenya, 3 National AIDS and STI Control Program (NASCOP), Ministry of Health, Nairobi, Kenya, 4 Department of Global Health, University of Washington School of Public Health and School of Medicine, Seattle, Washington, United States of America, 5 Centre for Clinical Research, Kenya Medical Research Institute (KEMRI), Nairobi, Kenya, 6 Support for Addictions Prevention and Treatment in Africa (SAPTA), Nairobi, Kenya, 7 Department of Epidemiology, University of Washington School of Public Health, Seattle, Washington, United States of America, 8 University of Washington Global Assistance Program-Kenya, Nairobi, Kenya

‡ LM and NTLB are co-senior authors on this work.
* mauranom@uw.edu

## Abstract

In Kenya, overdose remains a major public health concern with approximately 40% of persons who inject drugs (PWID) reporting personal overdoses. PWID living with HIV (PWID-LH) are particularly vulnerable to experiencing fatal and non-fatal overdoses because of the surrounding physical, social, economic, and political environments, which are not fully understood in Kenya. Through qualitative inquiry, this study characterizes Kenya's overdose risk environment. Participants were purposively recruited from a larger cohort study from September to December 2018 using the following inclusion criteria: HIV-positive, age ≥18 years, injected drugs in the last year, and completed cohort study visits. Semi-structured interviews explored experiences of personal and observed overdoses, including injection settings, sequence of events (e.g., pre-, during, and post-overdose), safety strategies, and treatment. Interviews were transcribed, translated (Swahili to English), reviewed, and analyzed thematically, applying a risk environment framework. Nearly all participants described personal and/or observed overdose experiences (96%) and heroin was the most frequently reported substance (79%). Overdose precursors included increased consumption, polysubstance use, recent incarceration, and rushed injections. There were also indications of female-specific precursors, including violence and accessing prefilled syringes within occupational settings. Overdose safety strategies included avoiding injecting alone, injecting drugs incrementally, assessing drug quality, and avoiding polysubstance use. Basic first-aid techniques and naloxone use were common treatment strategies; however, naloxone awareness was low (25%). Barriers to treatment included social network abandonment,

participant confidentiality, as we are unable to de-identify the dataset (i.e., qualitative transcripts), and we did not obtain consent for sharing the dataset. Our research team and institutional ethical review board understands the nature of the Plos One Mission wanting to provide data transparency to its readership; however, there are two reasons we are only able to share a restricted dataset at this time. First, this is a qualitative dataset (i.e., individual in-depth interviews) with participants sampled from a specific workplace in Nairobi, Kenya (i.e., peer educators working within a harm reduction site.) By providing the qualitative dataset, we run the risk of breaching participant confidentiality due to the nature of the data. More specifically, we are unable to de-identify the data as participants a) provided personal histories of their substance use and addiction and b) their introduction into working as a peer educator. Secondly, we did not disclose to participants during the study consent process that we would share the full dataset with journals. However, our research team and institution encourages collaboration and data transparency. While we are ethically unable to submit the full dataset, we would like to provide our study codebook, which includes code definitions, code frequencies, and application of the codes to the Plos One readership. Also, if there are inquiries about the data, readers should contact the study PI, Natasha Ludwig-Barron (Natasha.Ludwig@ucsf.edu) or the first author, Megan Maurano (mauranom@uw.edu). Please feel free to contact our University of Washington Institutional Review Board, Human Subjects Division at hsdinfo@uw.edu or +01-206-543-0098.

**Funding:** This research was funded by the National Institute on Drug Abuse (NIDA), with the primary investigation being the SHARP study (R01 DA043409, awarded to CF). The senior author was funded by an NIH Diversity Supplement (NIDA; R01 DA043409-S1, awarded to NLB) and the University of Washington Global Opportunities Health Fellowship (also awarded to NLB). The funders had no role in study design, data collection and analysis, decision to publish, or preparation of the manuscript.

**Competing interests:** The authors have declared that no competing interests exist.

police discrimination, medical stigma, fatalism/religiosity, medical and transportation costs, and limited access to treatment services. In Kenya, the overdose risk environment highlights the need for comprehensive overdose strategies that address the physical, social, economic, and political environments. Morbidity and mortality from overdose among PWID-LH could be reduced through overdose prevention initiatives that support harm reduction education, naloxone awareness, and access, destigmatization of PWID, and reforming punitive policies that criminalize PWID-LH.

# 1. Introduction

Overdose is the leading global cause of preventable drug-related death, and opioid-related overdose deaths are a global health crisis. By 2030, the number of persons who use drugs in sub-Saharan Africa is projected to increase by >50% [1] as a result of the expanding international drug trafficking routes [2, 3]. In Kenya, there was a rapid influx in the late 1990s and early 2000s of the easily injectable "white crest" heroin in coastal cities that catered to international tourists, supplanting the more commonly inhaled "brown sugar" heroin [2]. By 2018, an estimated 18,000 people in Kenya were injecting drugs, 19% of whom were living with HIV (compared to 5.6% in the general population) [4]. Overdose is a leading cause of mortality among PWID in Kenya, with approximately 40% of PWID reporting experiencing a personal overdose [4], which may be higher for PWID living with HIV [5].

Harm reduction strategies have historically focused on reducing the impacts of HIV among PWID [3]. However, the United Nations Committee on Narcotic Drugs [6] has called for public health agencies to incorporate more comprehensive overdose prevention and treatment strategies. Kenya's Ministry of Health developed a national protocol for treatment, which outlines recommendations for pharmacological treatment, psychosocial interventions, and aftercare support to address overdoses [7].

Harm reduction methods, such as opioid agonist treatments (OAT) and naloxone access, have been foundational in the global response to rising overdose risk [8] and are central to Kenya's public health agenda. In 2013, Kenya was one of the first countries in Africa to institute a national needle and syringe program to reduce HIV transmission among PWID, and introduced OAT, which is largely methadone, starting in 2014 [9, 10]. Despite these efforts, only 26% of PWID in Kenya are on OAT [4] and treatment options remain inaccessible or cost-prohibitive for many [11]. Timely characterization of the overdose risk environment and identification of the unique factors contributing to overdose risk amongst PWID living with HIV (PWID-LH) is greatly needed to improve harm reduction services in Kenya and throughout sub-Saharan Africa.

Using a risk environment framework provides a conceptual overview of how an individual can influence and be influenced by their surrounding physical, social, economic, and political environments, which sustain drug-related morbidity and mortality [12, 13]. Specifically, the risk environment framework proposed by Rhodes et al. (2002) provides insight to how drug-related harms, like overdose, are not solely attributable to individual behaviors, but rather upheld and perpetuated by an individual's surroundings [12]. Moreover, PWID-LH may possess additional risks for experiencing an overdose compared to individuals without HIV [5]. Interventions that address surrounding risk environments, rather than focusing on individuals-level risks, have the potential to reduce drug-related harms, as do interventions that address stigma toward people who use drugs [14–17]. However, the Kenya-specific overdose risk environment for PWID-LH has been largely overlooked.

In this study, we sought to characterize the overdose risk environment for PWID-LH receiving harm reduction services in Nairobi, Kenya in order to provide recommendations for adapting existing programs and policies that aim to reduce overdose risks. To do this, we used semi-structured interviews to explore PWID-LH (1) experiences preceding, during, or following a personal or observed overdose; (2) current safety strategies to reduce overdose risk, including access to naloxone treatment; and (3) overdose barriers and facilitators within the social, economic, physical, and political environments, with a particular focus on the healthcare and criminal justice systems.

## 2. Methods

### 2.1. Ethics statement

All study procedures and materials, including consent forms, were approved by the University of Washington Institutional Review Board (Seattle, WA, USA) and the Kenyatta National Hospital Ethical Review Committee (Nairobi, Kenya). In addition, all study materials and procedures were drafted with community partners and the oversight of government (MOH) personnel and each participant provide written informed consent in English or Swahili.

### 2.2. Parent study

This qualitative study was nested within a larger prospective cohort study, known as the *Study of HIV/Hepatitis C*, *APS*, *and Phylogenetics for PWID* (SHARP; NIH R01DA043409), which was conducted in Nairobi, Kilifi, and Mombasa counties in Kenya. The study procedures are outlined elsewhere [18]. Briefly, the study partnered with trusted community organizations including harm reduction organizations, syringe programs (NSP), medication-assisted treatment (MAT) centers, and public health centers that provided physical space for study visits. Study recruitment was conducted by trained peer educators who provide grassroots harm reduction services and, using a standard script encouraged PWID to screen and enroll at the nearest study site. Individuals were eligible if (1) ≥18 years of age; (2) injected drugs in the previous year; (3) living with HIV (confirmed using an OraSure Test); and (4) willing to provide contact information for sexual and injecting partners. Data collection began in March 2018 and included baseline and 6-month interviewer-administered questionnaires and bloodwork.

### 2.3. Qualitative study sampling and recruitment

For this qualitative study, we recruited participants from September to December 2018 who (1) completed their 6-month SHARP study appointment; (2) agreed to be contacted for future studies; and (3) attended one of the three Nairobi County study facilities. We applied purposive sampling techniques to oversample women, who represent 10–20% of PWID, and to ensure participants were geographically representative in order to capture a wide range of experiences [19, 20]. Of 79 participants eligible for the qualitative study, we stratified participants by sex, study facility, and self-reported suboptimal HIV care (e.g., had not attended an HIV clinic in more than 6 months and/or not taking HIV medication). Using a randomization package in R, we selected participants from each stratification category, prioritizing participants who represented two stratification categories. Potential participants were approached by harm reduction staff, who explained study procedures using a standard script and invited participants to complete an in-depth interview. In total, 28 participants were enrolled, with two declining to participate due to permanent and semi-permanent relocations, and four who were unable to be contacted.

All qualitative study procedures and materials were approved by the University of Washington Institutional Review Board (Seattle, WA, USA) and the Kenyatta National Hospital/University of Nairobi Ethical Review Committee (Nairobi, Kenya). Participants provided written informed consent in Swahili or English and were reimbursed 400 Ksh ($4 USD) for their time and transportation.

## 2.4. Data collection & management

Two bilingual (Swahili/English), bicultural, Masters-level interviewers with extensive experience working with PWID-LH conducted the screening, informed consent, and semi-structured in-depth interviews. Part I of the interview guide provided close-ended questions to assess demographics, drug use, HIV care, and drug treatment history. Part II provided open-ended questions to elicit participant experiences related to HIV risks and barriers to care. Provided that interview topics could potentially trigger OAT patients to relapse, care was taken in offering counseling and post-interview resources through our community partners.

Interviews ranged from 30–75 minutes in length and were audio-recorded, transcribed verbatim in Swahili, and translated into English. Quality control procedures included a second study staff member cross-checking the audio files and transcripts, and a third staff member reviewing the translation (Swahili to English) for accuracy. Following each interview, interviewers provided dictation notes to summarize the main interview topics and to capture participants' physical and mental conditions, which were also transcribed. ATLAS.ti Version 8 (Berlin, Germany) was used to manage, merge, and analyze the transcript data, including the interviewer notes, into one integrated database.

## 2.5. Data analysis

Data collection and analysis were conducted in parallel, with research team members and community partners meeting weekly to discuss the main study topics and emergent topics. While *a priori* research questions focused on HIV risks and barriers to care, early study team discussions highlighted participant concerns for fatal and non-fatal overdoses. Notably, the topic of overdose was of interest to our community partners, and following a literature review by the senior author (NLB), it was noted there was limited information on the topic of overdose within the region. Thus, additional probes were incorporated into the interview guide following the first five interviews to capture personal and observed overdose experiences, with the goal of characterizing overdose experiences and safety strategies. Data collection ceased when conceptual saturation was reached after 28 interviews; whereby additional interviews would not elicit new information on the topic of interest [21].

Coding processes included both inductive and deductive approaches. The first (MM) and senior (NLB) authors reviewed the literature and developed a preliminary code list (i.e., deductive approach). After reviewing interviewer notes, which summarized the topics of interest, MM conducted open-coding on 5 transcripts, highlighting emergent themes, which were organized as either safety measures or as a sequence of events: pre-, during- or post-overdose (i.e., inductive approach). Similar themes were merged as common or recurring concepts, which were organized into typologies and classification schemes through team member consensus [22]. Codebook management, including organizing codes and developing code definitions, parameters, and examples, was performed by MM. MM coded 5–7 transcripts weekly within Atlas.ti., which were reviewed by NLB and discussed at weekly team meetings. Disagreements on how codes were applied were resolved through team member discussion. This process was repeated until all 28 transcripts were coded.

**Table 1. Overdose (OD) themes and brief definitions.**

| *Themes* | *Brief Definition* |
|---|---|
| Experiences of OD | Personal and/or observed experiences following the ingestion or application of a drug or substance that leads to a toxic state or death. |
| OD Precursors | Events directly preceding an OD including taking too much of one substance, injecting unknown substances, polysubstance use, injecting following a period of abstinence, and rushed injections. |
| Safety Strategies | Methods used to prevent an OD, which included avoiding injecting alone, testing batches of drugs in group settings, using small amounts of drugs incrementally, and avoiding polysubstance use. |
| OD Treatment Strategies | Methods applied or witnessed following an OD including first aid techniques aiming to arouse, increase blood flow and oxygenation, and prevent aspiration. |
| Naloxone | Discussions related to the knowledge of, access to and use of Naloxone or other OD reversal medications. |
| Harmful strategies | Non-evidence-based strategies to reverse an OD including a) ingesting cleaning products to expel drugs via vomiting, b) submerging hands and feet in cold water to awaken a person, and c) injecting additional drugs, in an attempt to reverse an OD. |
| Barriers to Treatment | Described as social, political and economic factors such as a) police discrimination, b) medical stigma, c) fatalistic and/or religious views, d) medical and/or transport costs, e) limited access to care, and f) timeliness of OD treatment services. |
| Facilitators to Treatment | Though inconsistently applied, law enforcement agents providing options for drug treatment enrollment, rather than arrest. |

Analysis of the codes resulted in the framing of overdose risks through a) personal safety measures, b) sequence of overdose events, including pre-, during, and post-overdose experiences, c) treatment strategies, and d) treatment barriers, with Table 1 summarizing our themes with respective definitions. We distinguished between personal and second-hand experiences, as well as fatal vs. non-fatal overdoses, whenever possible. Moreover, we categorized risks according to a risk environment framework: physical, social, economic, or political risks. Overarching themes are presented alongside representative quotes, using pseudonyms to protect participant identities. In addition, community names and/or locations were omitted to prevent further stigmatization of vulnerable communities.

## 2.6. Inclusivity in global research

Additional information regarding the ethical, cultural, and scientific considerations specific to inclusivity in global research is included in the S1 Text.

## 3. Results

### 3.1. Sample characteristics

In total, 28 PWID-LH were enrolled in our qualitative study, with demographic and drug use characteristics presented in Table 2. Recruitment for the current study occurred at least 6-months after the parent study baseline assessment, which resulted in 14 participants being enrolled in OAT services that provide local daily methadone treatment within clinical settings. Twenty-three participants self-reported drug use in the previous year and 19 of those had injected drugs. Drug use included heroin (n = 22), marijuana (n = 18), cocaine (n = 6), and prescription drugs (n = 10), with 22 participants reporting polysubstance use (i.e., using more than one substance). In the following sections, we summarize participants' observed and lived overdose experiences, focusing on overdose precursors, safety methods used to prevent and

**Table 2. Demographics, substance use, and HIV care among qualitative participants.**

| Characteristics | Total<br>n = 28<br>n (%) |
|---|---|
| Sex | |
| Male | 12 (42.9%) |
| Female | 16 (57.1%) |
| Age in years, average (range) | 37.1 (18–56) |
| Alcohol use (weekly) | |
| None | 17 (61%) |
| Consumes ≥2 drinks | 7 (25%) |
| Excessive drinking[1] | 4 (14%) |
| Drug use (previous year) | |
| Any drug use[2] | 23 (82%) |
| Injection drug use (IDU) | 19 (68%) |
| Common drugs reported[3] | |
| Heroin | 22 (79%) |
| Cannabis/Bhang | 18 (64%) |
| Cocaine | 6 (21%) |
| Prescription drug misuse | 10 (36%) |
| Poly drug use | 22 (79%) |
| OAT services (i.e., current methadone use) | 14 (50%) |
| Overdose experience (lifetime)[4] | |
| Peer | 23 (82%) |
| Fatal | 14(61%) |
| Personal | 6 (21%) |

[1]Consumed 6 or more alcoholic drinks on one occasion weekly

[2]Smoked, snorted, inhaled or ingested any drugs in the previous year

[3]Any drugs consumed in the previous year

[4]Self-reported overdose experience

treat overdose, and barriers to enacting those methods. Overdose barriers and facilitators associated with each level of the risk environment framework are summarized in Table 3.

## 3.2. Experiences of overdose

Nearly all participants (27/28) had a firsthand experience with overdose. Twenty-three observed an overdose and/or knew someone who had overdosed, and six participants described personal overdose experiences. Fourteen (50%) witnessed fatal overdoses, and two participants were unsure or unable to determine the outcome of the observed overdose. Darweshi (M, 34 yrs), who has used drugs for more than 20 years describes seeing, *"very many [who] have died as a result of overdosing"*, including friends and family. The drugs being used at the time of the overdose were not always known. The most frequently cited overdose descriptors were fainting, being dizzy, not able to wake up, going limp, and falling down, which 12 participants described for both fatal and nonfatal overdoses.

The majority of participants described a sense of community, whereby PWID peers would help someone experiencing an overdose. Farashuu (F, 36 yrs) has never experienced an overdose but engages in polysubstance use that includes heroin, prescription opioids, and *chang'aa*

**Table 3. Barriers to OD prevention and treatment, and possible interventions identified in this study.**

| Barrier | Intervention | Target Population | Risk environment addressed | Example |
|---|---|---|---|---|
| Effective prevention and response | Educational programs | PWID | Social, Political | • Educational booklet<br>• Community events<br>• establishment of DCRs<br>• Good Samaritan Law |
| Police discrimination | Occupational campaign | Police | Social | • Harm reduction education<br>• Incentivizing treatment vs punitive approaches |
| Medical stigma | Educational programs | Healthcare providers and clinical staff | Social | • On-going educational seminars |
| Naloxone access | Provision of naloxone | PWID, incarcerated populations (upon release), community members, first responders | Economic & Political | • Naloxone distribution upon release from incarceration<br>• PWID access and training for personal use<br>• establishment of DCRs |
| Naloxone access | Training for chemists/pharmacists | Chemists/pharmacists | Political | • Naloxone at all pharmacies |
| Delays in treatment | Continuing Medical Education (CME) | Healthcare workers | Political | • Addiction medicine<br>• Overdose response protocols |
| Delays in treatment | Police Training | Police/First Responders | Political | • Overdose recognition and response protocols |

(i.e., homebrew). She explains that PWID cannot depend on *raya*, the general public, to help someone during an overdose, so PWID maintain trust and depend on one another:

> *"I always inject when we are three or two of us. So when things go wrong, the others will know how to help me. We only trust each other."*

> - Farashuu (F, 36 yrs)

While the majority of participants described a sense of community when an overdose occurred, four participants provided examples of when PWID peers abandoned someone experiencing an overdose. Abandonment was often driven by fear and the anticipated repercussions of being associated with an overdose. Kerubo (F, 28 yrs) engaged in polysubstance use of alcohol and intravenous heroin for more than 10 years. During that time, she witnessed several overdoses where:

> *"Some people run away because they feel they will be witnesses when one dies. But, as for me, I am used to it and [I am] always helping them."*

> *-Kerubo (F, 28 yrs)*

Specifically, PWID feared legal repercussions, having to tell a deceased peer's family and friends of their passing, having to reveal their own drug use and trauma associated with witnessing a fatal overdose.

### 3.3. Overdose precursors

Participants described several situations that led up to a personal and/or observed overdose, including taking too much of one substance, injecting unknown substances, polysubstance

use, injecting following a period of abstinence, and rushed injections. Rushed injections were associated with withdrawal symptoms, forced displacement during citywide clean-up efforts, and police presence. Various circumstances prevented PWID from confirming the identity or purity of what they injected. Three women discussed needing assistance injecting themselves and purchasing pre-filled syringes of unknown purity from illicit breweries where *chang'aa* (i.e., homebrew) is sold. Jimiyu (M, 41 yrs) has used heroin since the age of 17 and described multiple overdose experiences. His first overdose occurred within the first year of injecting drugs, where he injected too much heroin at once, and another time he describes polysubstance use of drinking alcohol and injecting heroin. Jimiyu describes his most recent overdose, which he attributes to his high opioid tolerance and forgetting to track his daily heroin intake:

> *"I felt that the steam had gone low and I even used to forget, I had forgotten, that I had taken it earlier and I go back to get it again. My brain is becoming slower and cannot cooperate as much as it used to."*

> - Jimiyu (M, 41 yrs)

This highlights overdose risk may be different for PWID with varying years of experience injecting drugs. Additionally, 11 participants described fears of injecting drugs of unknown purity or strength, with three participants describing fatal overdoses of PWID peers that resulted from substances the user did not procure themselves.

Similarly, individuals also faced a high risk of overdose when injecting heroin for the first time or following extended periods of abstinence, either through incarceration or via OAT. Two participants described overdoses resulting from unintentionally injecting too much of one drug, particularly when it was someone's first time injecting. Gathi (M, 56 yrs) and another participant highlighted former peers who experienced a fatal overdose after being released from prison and injecting what was a typical amount of drugs pre-incarceration:

> *"We were sentenced for 6 months and we swore, we will never smoke this thing again. So when we got to town every one of our friends were having a good time smoking bhang [cannabis]. We agreed to smoke and inject a piece [of heroin] each. That we buy one and divide in half for each of us. The problem is that we have gone so long without taking drugs while we were in prison. And we was using these ones from [drug purchasing site] and he drank alcohol and then went ahead and injected drugs, that's why [friend's name] died."*

> -Gathi (M, 56 yrs)

Forced drug abstinence during periods of incarceration, resulting in the loss of heroin tolerance, and polysubstance use are likely contributors to fatal overdoses in the region. Similar to Gathi's experience, seven other participants attributed personal overdose experiences to polysubstance use, both intentionally and unintentionally. Participants most often described injecting heroin combined with using alcohol, methadone, prescription medication, or unidentified substances. Though nearly all participants were aware of the overdose risk attributed to polysubstance use, half of the participants observed overdoses related to polysubstance use.

Participants described both political and physical factors that likely increased their risk of overdose through rushed injections. These included withdrawal symptoms or 'arosto', forced displacement driven by citywide clean-up efforts, crowded spaces, and police presence. Like several participants that experienced rushed injections because of 'arosto' or withdrawal symptoms, Darweshi (M, 41 yrs) worried that he would unintentionally inject too much or miss a vein because of his unstable hands while experiencing withdrawal symptoms. Naserian (F, 42

yrs), who engaged in sex work to support herself and to pay for heroin, described having to purchase and inject her drugs quickly within crowded den settings, which are outdoor public spaces where drugs are purchased, sold and consumed, in order to avoid harassment from PWID peers and police:

> *"When you go to [location] you can hurt yourself even further. People are coming in groups, so you want to finish before this other group comes. . . And the municipal police made it very difficult at [location] and harassed us."*
>
> - Naserian (F, 42 yrs)

Naserian described her preferred den as having a constant police presence; however, seven other participants described police presence as being inconsistent, often coinciding with city-wide clean-up efforts initiated by government officials. Absko (M, 50 yrs), who has been injecting drugs for more than 20 years, described the changes in police presence that force many PWID to rush their injections:

> *"The government has distracted our peace. We are no longer free as we used to be earlier on. The government comes there, and if they find you, it is war. They can even take you to the police station, and yes, they beat you up, seriously."*
>
> –Absko (M, 50 yrs)

Similar to Absko's experience, other participants also described injecting their drugs quickly to avoid police conflict and physical abuse. Injecting quickly increases overdose risk by preventing PWID from injecting incrementally, a harm reduction safety measure applied when drug contents or quality is unknown. As in Naserian's case, women may have additional social risks, such as rushing injections in order to avoid physical violence and sexual harassment by PWID male peers and police.

In summary, the precursors of overdose amongst PWID-LH in Kenya included (a) purchasing pre-filled syringes, (b) depending on peers to purchase drugs, (c) recent drug abstinence, (d) using substances of unknown contents or purity, (e) polysubstance use, and (f) rushed injections due to withdrawal symptoms, crowded injection settings, government-initiated displacement, and police presence. In addition, women may face additional overdose risks that are not experienced by men. Women who engage in sex work and injection drug use may rush injections due to threats of physical and sexual violence perpetrated by male peers or police. Furthermore, while purchasing pre-filled syringes may be uncommon, this may occur within specific physical environments like illicit breweries and/or sex work venues, which are frequented by women who inject drugs.

### 3.4. Overdose safety strategies

Nearly all participants (n = 24) described ways to prevent overdoses, including relying on PWID peers, testing batches of drugs in group settings, using small amounts of drugs incrementally, and avoiding polysubstance use. Avoiding injecting drugs alone was the most common safety strategy (n = 11), particularly when participants feared drugs might be adulterated or of unknown purity or strength. Participants trusted their peers would be able to detect overdose symptoms and attempt to intervene when an overdose occurred. Nearly all participants described helping another PWID during an overdose by placing the individual in a recovery position, calling for help and/or treatment, or preventing the affected person from using more substances.

For at least three participants, injecting drugs within a group setting had the additional benefit of being able to see how others reacted to the drugs being injected, before injecting themselves. Naserian (F, 42 yrs) has never trusted the drug dealers who say the drugs are "pure," or without adulterants. She waits to see someone else inject as a way of "testing the drugs" before she injects herself. Seeing the reaction of a peer immediately following an injection offered Naserian a sense of security that she could inject without overdosing. Thus, there may be multiple harm reduction benefits to injecting drugs within group settings.

Another safety strategy included using drugs incrementally to prevent unintentionally injecting too much at once, which was described by seven participants. Two participants, Gathi and Kioko, described different approaches to implementing this strategy. Gathi (M, 56 yrs) had been taught by a PWID peer to never inject all of his drugs at once. Instead, he takes only a fraction of what he intends to inject and injects multiple times as long as there are no adverse outcomes. Kioko (M, 54 yrs), who has been using heroin for almost 30 years, does not trust himself to use his drugs in incremental, small amounts. He describes his daily routine of procuring and injecting his drugs at least twice per day:

> *Kioko*: *I buy one dose and when I need another one, I go back and buy it again.*
>
> *Interviewer*: *You can't buy all of it at once?*
>
> *Kioko*: *You cannot buy and keep it.*
>
> *Interviewer*: *Why?*
>
> *Kioko*: *When you keep it, you will get the urge to use it and you can use all of it and overdose. So usually, I buy like two [doses] and then wait for some time before going back again to buy some more.*
>
> -Kioko (M, 54 yrs)

Incremental, small-batch injections are established harm reduction techniques that are disseminated through peer educator networks in Nairobi and other major cities. Understanding PWID applied safety approaches, like Gathi's and Kioko's, offers opportunities to tailor overdose prevention programs to local PWID.

Nearly half of the participants (n = 13) described not mixing substances and/or using certain drugs before others as a strategy to avoid overdose, crediting harm reduction peer educators and lived overdose experiences for their overdose prevention knowledge. Hanuni (F, 45 yrs) works and lives at an illicit brewery, where alcohol and drug access are abundant. She and two other participants endorsed safety strategies of injecting their heroin before drinking any alcohol, which are both respiratory depressants. By injecting their drugs first, followed by slowly drinking alcohol, participants felt they could reduce their overdose risk by closely monitoring symptoms. Similarly, nine participants credited peer educators, peers, and OAT clinical staff who educated them on the increased overdose risk when combining heroin and methadone. Findings point to the recently implemented educational campaigns that highlight overdose risks of taking both methadone and heroin are reaching PWID communities.

In summary, we identified multiple harm reduction techniques in practice, including injecting drugs incrementally, injecting in group settings, and avoiding mixing substances. In addition, dissemination of harm reduction evidence-based strategies has been successful in reaching PWID-LH, but they may have limited effectiveness without additional support.

### 3.5. Overdose treatment strategies

Seventeen participants described several ways to treat someone experiencing an overdose, many in alignment with Harm Reduction International's recommendations for responding to an opioid overdose [23]. However, 11 participants were not aware of naloxone as a treatment for an opioid-related overdose, highlighting an opportunity to provide education on harm reduction strategies.

Six participants described first aid techniques for PWID peers during an overdose, which aimed to rouse, increase blood flow and oxygenation, and prevent aspiration. Nekesa (F, 30 yrs), who is currently on methadone, but was previously injecting heroin five times a day, described placing a PWID peer in the recovery position:

> "It was [an] overdose and because I know how people like that are handled [in the hospitals], I put him to lie down on his side, unfolded the legs and straightened the hands. We did all that but after a few minutes he died."
>
> -Nekesa (F, 30 yrs)

Similar to Nekesa, participants described laying an affected person down to maintain open airways, unbuttoning clothing, and removing belts and shoes to promote cooling and circulation, which are considered basic first-aid techniques. While these techniques have limited ability to reverse an overdose, they highlight that many overdoses occur in the presence of PWID peers who are prepared to intervene but in need of more effective tools.

Of concern, there were two participants who described strategies that could cause additional harm when someone is experiencing an overdose. Anyango (F, 30 yrs), who engages in sex work to support herself and her family, described a personal overdose:

> "My friend came and gave me the bad one [adulterated drugs]. The one [drug] they give people to pass out so that they can steal from them. I blacked out and foam was coming from my mouth, so I was given 'omo' [cleaning product] and I vomited. Then I was given milk with sugar. They added a lot of sugar and then I came back to normal. I slept and then felt better."
>
> -Anyango (F, 30 yrs)

Using cleaning products or detergent to expel drugs via vomiting can cause additional health-related harm. Other ineffective or potentially dangerous responses implemented by PWID included a) putting hands and feet in cold water and b) injecting additional drugs, in an attempt to reverse an overdose (e.g., stimulants were injected following an opioid-related overdose and visa-versa.) Adapting harm reduction policy standards and educating PWID on evidence-based overdose treatment strategies has the ability to reduce overdose fatalities.

Knowledge of effective overdose reversal medications was limited, with seven participants (25%) aware of naloxone or any medication used to treat opioid overdose, regardless of name. Participants who were aware of naloxone had witnessed its application by a harm reduction specialist. Participants described naloxone access through local harm reduction facilities, peer educators, outreach workers, hospitals, and chemists (i.e., pharmacists); however, none of the participants had naloxone readily available. In Nairobi, participants like Gacocki (M, 30 years) were relieved to learn about naloxone, though its access was limited and not for immediate use. Naloxone is available in densely populated injection settings like dens that contain 20 to 300 PWID and dealers, through peer educators, pharmacies and clinics known to treated PWID patients. However, these points of access may be of limited utility as harm reduction

facilities, peer educators, clinics, and pharmacies have limited service hours in the evenings and on weekends.

In summary, participants described using both effective and ineffective first aid techniques and therapeutic medications to respond to overdose, highlighting the opportunity to better train PWID as "first responders." However, naloxone knowledge and access were limited.

### 3.6. Barriers to seeking overdose treatment

Participants described several social, economic, and political barriers to treating an overdose. Generally, law enforcement agents were seen as barriers to accessing overdose treatment by refusing to respond to an overdose. Additionally, 20 participants described physical violence perpetrated by police. Conversely, two participants described instances where police officers took less punitive approaches to drug possession charges by offering transport to harm reduction and OAT facilities, which is an evidence-based harm reduction strategy. Similarly, certain clinics were seen as barriers when PWID experienced an overdose. Several participants like Naserian (F, 42 years) explained, *"Nooo. For [overdose] people don't go to the hospital,"* for fear of being arrested, experiencing drug-related stigma that could result in poor treatment by clinicians, or both. As such, participants described police discrimination and medical stigma as barriers to seeking overdose treatment.

In addition, fatalism and religious beliefs surrounding overdose was an emergent theme described by two participants. Absko (M, 50 years), who injects heroin and smokes bhang at least three times per day, explained that he follows harm reduction recommendations, but regardless of the medical intervention, PWID have predetermined destinies:

> *"We will pull him aside and lay him in recovery position, if the needle is still in the body we will remove it, give him some air, and if he was meant to survive by God's help, he will wake up. But if his day-to-day had reached, he will just die there as we watch."*

> - Absko (M, 50 years)

While religiosity and belief systems could be viewed as barriers to overdose interventions, this emergent finding may offer opportunities and insight into developing culturally-tailored harm reduction programs.

Political and economic barriers included treatment access and costs associated with clinical care and transportation, which were discussed by four participants. For instance, Chepkirui (F, 32) experienced overdose symptoms after injecting heroin, but resisted seeking medical attention due to her inability to pay; however, Chepkirui's peers arranged for her transport to a local clinic known to treat PWID at no cost. More often, transportation costs were a barrier for PWID-LH who lived outside of city limits and for those needing to attend multiple clinic facilities (e.g., OAT, maternal health). Lack of widespread access to care and treatment creates logistical barriers for PWID-LH experiencing overdose. Two participants explained that the process and time it takes to access medical aid has led to PWID-LH experiencing fatal overdoses. Furthermore, access was dependent on when an overdose occurred as many facilities were not open or provided limited hours during evenings and weekends.

In summary, barriers to care include police discrimination, medical stigma, fatalistic and/or religious views, as well as more conventional barriers to care, like medical costs, transportation costs, access to care, and timeliness of overdose treatment services.

## 4. Discussion

Using qualitative inquiry, our analysis is among the first to characterize the overdose risk environment among PWID-LH in Nairobi, Kenya. Nearly all participants had observed or personally experienced an overdose, underscoring the regional prevalence, and likely underreporting, of overdoses. Dependence on PWID peers for support and assistance during an overdose was common. However, trauma, fear, and anticipated legal repercussions contributed to cases where peers abandoned an individual experiencing an overdose. Overdose precursors included increased consumption, polysubstance use, injecting drugs following periods of abstinence, and rushed injections. Female-specific risks included access to prefilled syringes within illicit breweries and sex work venues, and rushed injections to avoid harassment and violence by male PWID and police. Participants are implementing evidence-based harm reduction practices that minimize overdose risks, including (a) avoiding injecting drugs alone, (b) injecting small amounts of drugs incrementally, (c) avoiding polysubstance use, (d) providing first aid, and (e) attempting to access naloxone.

Based on our interviews with PWID-LH, policy and programmatic recommendations to reduce fatal and non-fatal overdoses include (a) expanding access to overdose treatment services (i.e., naloxone) and addressing treatment misconceptions, (b) enacting policies that address police discrimination and reduce medical stigma towards PWID, (c) advocating for policies that allow PWID peers to assist when an overdose occurs (i.e., Good Samaritan Laws), and (d) addressing conventional barriers to treatment (e.g., medical and transportation costs, treatment access, and timeliness of treatment services.)

Our findings align with global overdose assessments, as several factors associated with overdose in Kenya have previously been described within other PWID communities [5, 24]. These include the resumption of drug use after periods of abstinence, polysubstance use, and use of drugs with unverified purity and potency. In particular, polysubstance use is a key issue in Kenya. Educational initiatives within harm reduction agencies, as well as policies to inform OAT patients about risks associated with polysubstance use and overdose, are ongoing [7]. Additionally, we identified social and political pressures that increase overdose risk, such as politically-driven, citywide clean-up efforts that cause PWID to rush their injections. Our findings echo those in Tanzania where polysubstance use–especially the combination of injection drugs with alcohol consumption–was associated with forced displacement and rushed injections [25, 26]. Beyond Kenya, there has been a rise in synthetic opioids, with increased potency and greater risk of fatal overdose, mainly fentanyl and xylazine, in drug markets [27–29]. Timely efforts are needed to increase overdose awareness and additional harm reduction investments are needed to prevent overdoses in an evolving drug market [30–32].

Participants described several low-cost overdose prevention and treatment strategies that can be enhanced and expanded in Kenya. Several participants reported injecting small amounts of drugs incrementally, avoiding polysubstance use, avoiding injecting alone, and testing drug batches prior to use, strategies recommended by the National Harm Reduction Coalition [33]. Most safety strategies were communicated through the social networks of PWID peers and peer educators, who are former PWID, which could be leveraged for future overdose prevention interventions. Participants also described basic first aid techniques to reduce the risk of an overdose [33]. Overdose prevention campaigns could be more widespread among PWID communities and work towards (a) dispelling misinformation around using household cleaning products or injecting drugs thought to counteract overdoses, which can increase overdose risk and cause additional complications, and (b) integrating overdose and HIV services for PWID-LH.

Stigma in multiple forms was associated with increased of overdose. Several participants associated law enforcement with displacement, trauma, and physical harm, which undermines efforts to improve harm reduction for PWID-LH. Medical stigma also led to poor outcomes, discouraging PWID from seeking care, and delaying care by limiting facilities PWID could go. Recently, Nairobi-based harm reduction organizations have partnered with local police departments to provide training on medicalized approaches, whereby PWID arrested for drug possession are given the option of jail or OAT services. This strategy has been successful in reducing violence and drug-related stigma in other settings, and has the benefit of reducing civil costs associated with incarceration [34]. Thus, supporting policies that view law enforcement agents as public health liaisons can improve the health and well-being of PWID-LH and reduce overdoses through community-based harm reduction interventions. Similarly, protective policies that encourage collaboration between government and social service agencies can reduce community displacement that increases overdose risk. Several policies that help destigmatize drug use while simultaneously providing safer use environments, have been effective at reducing opioid overdoses in other countries, including Drug Consumption Rooms (DCRs), and Good Samaritan laws, which could be introduced in Kenya. DCRs are locations where illicit drug consumption can be monitored by trained staff and naloxone can be provided on-site. DCRs allow PWID to inject drugs without being rushed or harassed and to receive assistance if they overdose. DCRs may also facilitate drug treatment, access to health services, and cessation of drug injecting [35]. Additionally, Kenya could benefit from Good Samaritan laws that aim to reduce overdose mortality by eliminating punitive measures when PWID report the overdose of a peer [36]. Global policies are shifting away from punitive sanctions that exhaust incarceration facilities and are moving towards decriminalization of drug use and possession, as well as integrated public health approaches that recognize drug addiction as a mental health issue. Such policies could be successful in Kenya, given their commitment to harm reduction strategies and reducing overdose-related morbidity and mortality.

Barriers to overdose harm reduction included low awareness about and access to opioid overdose treatment. Several countries are conducting overdose prevention and anti-stigma campaigns that have reduced overdose mortality, [37, 38] but their benefits are likely limited without additional investment in effective overdose treatments [39, 40]. Naloxone was added to Kenya's List of Essential Medications in 2003 [41] but is often limited by distribution, access points, and short shelf life [42]. During COVID-19 lockdown periods, there were localized African movements that advocated for policy changes towards widespread distribution of naloxone, which had varying success [43]. Similar to our study population, several PWID in South Africa were unaware of naloxone (>60%), but once described, more than 70% of PWID participants said they would feel comfortable carrying naloxone with them [44]. A review of take-home naloxone studies in low and middle-income countries suggests that people who use drugs benefit from training on overdose response, including how to deliver naloxone, and found little to no indication that widespread availability of naloxone increases opioid use [42]. Though naloxone is available through harm reduction agencies, medical facilities, and pharmacists, additional advocacy and government support are needed to allow PWID to obtain and carry naloxone so that it is immediately accessible during an overdose [33]. Naloxone dissemination interventions within incarcerated populations following their release have been effective in reducing overdoses elsewhere and could be effective within Kenya [45]. By leveraging existing resources, including the social networks and the sense of community reported by participants, public health officials may optimize the health benefits of overdose prevention strategies, response education, and opioid overdose treatment. We believe that reducing overdoses and overdose deaths in Kenya requires pairing social (e.g., educational campaigns) and political interventions (e.g., policies that improve access to naloxone).

### 4.1. Limitations and strengths

Findings were limited to the experiences of PWID-LH recruited from two harm reduction facilities and one OAT clinic in Nairobi, Kenya, which may not be applicable to other PWID settings. Recall bias may have distorted the accuracy of personal and observed experiences of overdose; however, several accounts provided foundational details on overdose experiences, safety strategies, treatment knowledge, and barriers to treatment. While local research staff with extensive experience working with PWID-LH conducted interviews, transcribed and translated interviews, we acknowledge that translated terms and concepts may not exist in the English language, as they do in Swahili. Also, while the cause of an overdose was rarely explicit, most descriptions were consistent with opioid-related overdoses, which have been more difficult to study in other settings. Furthermore, Kenya's drug market is constantly shifting, thus, overdose type, risk, and effective interventions are likely to change over time. Finally, the study presented here is limited to characterizing the environment experienced by PWID-LH, which may be impacted by intersecting vulnerabilities (e.g., mental health, HIV-related stigma, etc.). Future studies should consider the experiences of clinicians, local policymakers, and law enforcement agents to broaden our understanding of the overdose risk environment.

## 5. Conclusions

In summary, we explored environmental risks surrounding personal and observed overdose experiences, including both fatal and non-fatal overdoses, among PWID-LH in Nairobi, Kenya. This study expands our understanding of Kenya's overdose risk environment, including overdose precursors, safety strategies, treatment, and barriers to care. PWID-LH are largely dependent on peers and grassroots harm reduction services for overdose prevention and treatment; whereas, law enforcement agents and clinical staff were perceived barriers to implementing harm reduction strategies and accessing overdose treatment. Thus short-term, community-level interventions that leverage social networks may provide immediate overdose reductions through peer-supported educational campaigns, with policy-level interventions that expand treatment access and address discrimination providing sustainable overdose outcomes. In closing, both fatal and non-fatal overdoses are preventable health outcomes that can and should be supported through concerted community and political actions.

## Supporting information

**S1 Text. Inclusivity in global research questionnaire.**
(DOCX)

## Acknowledgments

We would like to thank the team at SAPTA who made this study possible. We thank the staff who facilitated the study and the peer educators who volunteered to participate in the interviews.

## Author Contributions

**Conceptualization:** Esther Gitau, Brandon L. Guthrie, Aliza Monroe-Wise, William Sinkele, Carey Farquhar, Loice Mbogo, Natasha T. Ludwig-Barron.

**Data curation:** Mercy Apiyo Owuor, Betsy Sambai, Loice Mbogo, Natasha T. Ludwig-Barron.

**Formal analysis:** Aliza Monroe-Wise, Loice Mbogo, Natasha T. Ludwig-Barron.

**Funding acquisition:** Brandon L. Guthrie, Aliza Monroe-Wise, Carey Farquhar.

**Investigation:** Megan Maurano, David Bukusi, Sarah Masyuko, Rose Bosire, Esther Gitau, Mercy Apiyo Owuor, Betsy Sambai, Loice Mbogo.

**Methodology:** Esther Gitau, Aliza Monroe-Wise.

**Project administration:** David Bukusi, Rose Bosire, Esther Gitau, Aliza Monroe-Wise, Loice Mbogo.

**Supervision:** David Bukusi, Sarah Masyuko, Esther Gitau, Aliza Monroe-Wise, Loice Mbogo, Natasha T. Ludwig-Barron.

**Writing – original draft:** Megan Maurano, Natasha T. Ludwig-Barron.

**Writing – review & editing:** Megan Maurano, David Bukusi, Sarah Masyuko, Brandon L. Guthrie, Helgar Musyoki, Hanley Kingston, Carey Farquhar, Loice Mbogo, Natasha T. Ludwig-Barron.

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
