## [Decision Letter · Decision Letter 0]

16 Apr 2024

PGPH-D-23-02150

“We only trust each other”: A qualitative study exploring the overdose risk environment among persons who inject drugs living with HIV in Nairobi, Kenya

Dear Dr. Maurano,

Thank you for submitting your manuscript to PLOS Global Public Health. After careful consideration, we feel that it has merit but does not fully meet PLOS Global Public Health’s publication criteria as it currently stands. Therefore, we invite you to submit a revised version of the manuscript that addresses the points raised during the review process.

The first author and co-senior authors are all affiliated with US institutions. As this is a study in Kenya, senior authorship should include a Kenyan author or an author affiliated with one of the Kenyan institutions. 

Which author is affiliated with [i) Institute of Public Health Genetics, School of Public Health, University of Washington, UW Box 33 # 351619, Seattle, WA, 98195, USA] as this is not listed next to any of the authors names? 

Please see the comments from reviewers below, please make the changes as listed for acceptance. Please note from reviewer 2 comments, from my perspective as editor, the inclusion of the interview script is not required for acceptance but a summary figure with the main themes as sub-themes would be helpful.

We look forward to receiving your revised manuscript.

Kind regards,

Padmasayee Papineni, MD

Academic Editor

Journal Requirements:

2. Please include a complete copy of PLOS’ questionnaire on inclusivity in global research in your revised manuscript. Our policy for research in this area aims to improve transparency in the reporting of research performed outside of researchers’ own country or community. The policy applies to researchers who have travelled to a different country to conduct research, research with Indigenous populations or their lands, and research on cultural artefacts. The questionnaire can also be requested at the journal’s discretion for any other submissions, even if these conditions are not met.  Please find more information on the policy and a link to download a blank copy of the questionnaire here: https://journals.plos.org/globalpublichealth/s/best-practices-in-research-reporting. Please upload a completed version of your questionnaire as Supporting Information when you resubmit your manuscript.

3. Please amend your detailed Financial Disclosure statement. This is published with the article. It must therefore be completed in full sentences and contain the exact wording you wish to be published.

If you did not receive any funding for this study, please simply state: “The authors received no specific funding for this work.

Additional Editor Comments (if provided):

Reviewers' comments:

Reviewer's Responses to Questions

**Comments to the Author**

1. Does this manuscript meet PLOS Global Public Health’s publication criteria? Is the manuscript technically sound, and do the data support the conclusions? The manuscript must describe methodologically and ethically rigorous research with conclusions that are appropriately drawn based on the data presented.

Reviewer #1: Yes

Reviewer #2: Yes

2. Has the statistical analysis been performed appropriately and rigorously?

Reviewer #1: N/A

Reviewer #2: Yes

3. Have the authors made all data underlying the findings in their manuscript fully available (please refer to the Data Availability Statement at the start of the manuscript PDF file)?

Reviewer #1: Yes

Reviewer #2: Yes

4. Is the manuscript presented in an intelligible fashion and written in standard English?

Reviewer #1: Yes

Reviewer #2: Yes

5. Review Comments to the Author

Reviewer #1: Overall the paper reads well and is a good addition to the evidence available in kenya and the region regarding overdose occurence, prevention and response.

Consider re-structuring the sentence line 87-89 on opioid overdose being the leading cause of mortality among PWID in Kenya. The referenced article only indicates the percentage of opioid overdose reported (40% in Kenya), that it is a leading cause (from a meta-analysis)

Line 153 make a correction on the reimbursement amount, as the Ksh value has a dollar sign on it as well

269-273- the quote is not indicative of overdose, but rather the effect of an unknown substance

415- were there examples of stimulants injected that were provided? There is need to mention these

427 grammatical correction

overall, were there specific shared experiences of heroin methadone polyuse and overdose?

Reviewer #2: This paper aims to explore the characteristics of overdose risk environments for people who inject drugs and are living with HIV in Kenya, with the goal of identifying recommendations to improve existing policies and services for reducing overdose risks. This qualitative study is well-conducted. Below, authors can find my comments aimed at improving the paper:

pg 3: keywords - As PWID is used as a keyword for indexing, it might be better to have it listed separately, outside of brackets.

pg 4: In the introduction's first paragraph, you provide information about different drugs such as opioids or heroin (referred to as white crest and brown sugar). However, switching between different drugs may make it somewhat challenging for the reader to follow. Authors may consider focusing on one type of drug to effectively highlight the problem.

pg 5: You've hinted at reasons for choosing the risk environment framework, but it could be clearer for the reader why this framework was chosen for your study. Providing more explicit rationale for selecting this framework would enhance the clarity and understanding of your research approach.

pg 6: can you provide the reasons for declining to participate if known?

Pg 7: “OAT patients” If this abbreviation is not clarified before please provide its full meaning within the article

It would be beneficial to include the interview script in the paper for the audience's reference.

Additionally, could you summarise the main themes and sub-themes in a figure to help readers easily identify them?

If the interview scripts were translated into English, it would be important to acknowledge this as a potential limitation.

6. PLOS authors have the option to publish the peer review history of their article (what does this mean?). If published, this will include your full peer review and any attached files.

**Do you want your identity to be public for this peer review?** For information about this choice, including consent withdrawal, please see our Privacy Policy.

Reviewer #1: No

Reviewer #2: No

---

## [Editor Report · Decision Letter 1]

13 Jun 2024

“We only trust each other”: A qualitative study exploring the overdose risk environment among persons who inject drugs living with HIV in Nairobi, Kenya

PGPH-D-23-02150R1

Dear Dr Maurano,

We are pleased to inform you that your manuscript '“We only trust each other”: A qualitative study exploring the overdose risk environment among persons who inject drugs living with HIV in Nairobi, Kenya' has been provisionally accepted for publication in PLOS Global Public Health.

Best regards,

Padmasayee Papineni, MD

Academic Editor

Thank you for the revised manuscript and for the explanation of the changes made. I can now confirm that we accept the manuscript for publication.